# Spinal signalling of C-fiber mediated pleasant touch in humans

**Andrew G Marshall[1,2,3]\*, Manohar L Sharma[3], Kate Marley[4], Hakan Olausson[4,5,6], Francis P McGlone[2,7]**

[1]Institute of Aging and Chronic Disease, University of Liverpool, Liverpool, United Kingdom; [2]School of Natural Sciences and Psychology, Liverpool John Moores University, Liverpool, United Kingdom; [3]Department of Pain Medicine, Walton Centre NHS Foundation Trust, Liverpool, United Kingdom; [4]Specialist Palliative Care Team, University Hospital Aintree, Liverpool, United Kingdom; [5]Center for Social and Affective Neuroscience, Linköping University, Linköping, Sweden; [6]Department of Clinical Neurophysiology, Linköping University Hospital, Linköping, Sweden; [7]Institute of Psychology, Health and Society, University of Liverpool, Liverpool, United Kingdom

**Abstract** C-tactile afferents form a distinct channel that encodes pleasant tactile stimulation. Prevailing views indicate they project, as with other unmyelinated afferents, in lamina I-spinothalamic pathways. However, we found that spinothalamic ablation in humans, whilst profoundly impairing pain, temperature and itch, had no effect on pleasant touch perception. Only discriminative touch deficits were seen. These findings preclude privileged C-tactile-lamina I-spinothalamic projections and imply integrated hedonic and discriminative spinal processing from the body.

**\*For correspondence:**
andrew.marshall@liverpool.ac.uk

**Competing interests:** The authors declare that no competing interests exist.

## Introduction

There are several aspects of touch. In addition to a well-defined discriminative role, touch has an affective dimension of fundamental importance to physical, emotional and social well-being, both developmentally and throughout life (*McGlone et al., 2014*). C-tactile afferents, a subclass of unmyelinated low threshold mechanosensitive C-fibers innervating human hairy skin, are strongly implicated as the neurobiological substrate subserving the affective and rewarding properties of touch (*McGlone et al., 2014*).

C-tactile afferents have slow conduction velocities ($\sim$1 m s$^{-1}$) which, along with other neurophysiological properties such as fatigue to repeated stimulation, makes them poorly suited for tactile discrimination (*Vallbo et al., 1999*; *Olausson et al., 2010*; *McGlone et al., 2014*). Instead, microneurography and psychophysical investigations indicate that C-tactile afferents preferentially respond to tactile velocities and forces typical of a gentle caress (*Löken et al., 2009*; *Ackerley et al., 2014a*) with peak firing rates that positively correlate with perceived touch pleasantness (*Löken et al., 2009*).

In keeping with a role in signalling the affective aspects of touch, selective C-tactile stimulation activates contralateral posterior insula cortex (*Olausson et al., 2002*; *Olausson et al., 2008*), a region considered a gateway for sensory systems to emotional cortical areas (*Craig, 2008*), but not somatosensory areas S1 and S2 (*Olausson et al., 2002*; *Olausson et al., 2008*). In addition, patients with ischemic stroke affecting the posterior contralateral opercular-insular cortex demonstrate impairments in the perception of C-tactile optimal touch (*Kirsch et al., 2019*). Likewise, posterior insula activation is not modulated by C-tactile optimal stimulation in individuals with congenital

C-fiber denervation (*Morrison et al., 2011*). C-tactile-mediated affective touch pathways are, therefore, proposed to diverge from the Aβ low threshold mechanoreceptor afferent dorsal column/medial-lemniscal discriminative touch stream and form a distinct coding channel projecting primarily to emotional rather than classical somatosensory cortical regions (*Craig, 2002*; *Morrison et al., 2010*; *McGlone et al., 2014*).

The major somatosensory input into primate dorsal posterior insular cortex arise from the posterior ventral medial nucleus of thalamus (*Craig et al., 1994*; *Craig and Zhang, 2006*). Spinal inputs to this thalamic relay derive, almost exclusively, from projection cells in dorsal horn lamina I via the spinothalamic tract (*Craig and Zhang, 2006*). The central terminals of C-low threshold mechanosensitive receptor (C-LTMR) afferents, the animal equivalent of C-tactile afferents, arborise in laminae II/IIIi of the spinal cord dorsal horn (*Light and Perl, 1979*; *Sugiura, 1996*; *Li et al., 2011*; *Abraira and Ginty, 2013*; *Larsson and Broman, 2019*). Lamina II cells activated by C-LTMR afferents arborise in lamina I (*Lu and Perl, 2005*; *Maxwell et al., 2007*; *Lu et al., 2013*) where they can contact projection neurons (*Lu et al., 2013*).

Thus, the 'dual pathway' model of discriminative and emotional touch predicts that signals arising from C-tactile activation diverge from dorsal column-bound Aβ inputs to ascend alongside other small-diameter primary afferent modalities in the lamina I spinothalamic pathway. Accordingly, disruption of the spinothalamic tract, which lies within the anterolateral funiculus of the spinal cord, would, in addition to causing contralateral deficits in classical spinothalamic modalities of pain, temperature and itch, be predicted to induce alterations in affective but not discriminative touch domains. To test this prediction the effects of targeted spinothalamic tract ablation on discriminative and affective touch were investigated in patients undergoing anterolateral cordotomy to treat refractory unilateral cancer-related pain.

## Results

Assessment of noxious and innocuous temperature, itch and noxious mechanical sensation as well as discriminative and affective aspects of touch were performed on the pain-affected and unaffected sides in 19 patients undergoing anterolateral cordotomy. All sensory testing was performed on hairy skin of the dorsal forearm distant to the sites of clinical pain. No patient had pre-existing neurological deficits in the area of testing. The cordotomy was performed percutaneously at cervical level C1/C2 on the side contralateral to clinical pain. A cordotomy electrode was inserted under X-ray guidance in to the anterolateral funiculus of the spinal cord (*Figure 1a and b*). Lesioning was performed using a radiofrequency current to produce heat-induced lesions targeting the spinothalamic tract (for clinical and procedural-related information see Materials and methods and *Supplementary file 1*). The pre-test, post-test design resulted in four conditions; pre-cordotomy pain-affected, pre-cordotomy control, post-cordotomy pain-affected and post-cordotomy control.

As expected, anterolateral cordotomy induced clear-cut contralateral deficits in canonical spinothalamic modalities: there was striking amelioration of clinical pain (*Supplementary file 1*); perceptual thresholds for innocuous temperature and thermal pain were markedly elevated (Related-Samples Wilcoxon Signed Rank Test all p<0.0005) (*Figure 1c and d* and *Supplementary file 2a*) and in the majority of patients thermal sensibility was abolished; Cowhage-induced itch was abolished (*Supplementary file 2a*). In contrast tactile acuity and graphesthesia were unchanged (*Supplementary file 2a*). These findings, therefore, confirm marked cordotomy-induced disruption of lamina I spinothalamic pathways.

The pleasant aspects of touch were evaluated using structured psychophysical assessments based on characteristic C-tactile stimulus-response properties. C-tactile afferents respond optimally to gentle skin stroking and display peak firing rates to stroking stimuli delivered with velocities of ~3 cm s$^{-1}$ (*Löken et al., 2009*; *Ackerley et al., 2014a*). The resulting inverted U-shaped relationship of the neural response to brushing velocity is, critically, matched by subjective ratings of touch pleasantness (*Löken et al., 2009*; *Ackerley et al., 2014a*). Correspondingly, pre-cordotomy visual analogue scale (VAS) ratings for touch pleasantness to gentle brushing stimuli were greater to stroking at 3 cm s$^{-1}$ than at 0.3 and 30 cm s$^{-1}$ (*Figure 2a* - d). However, cordotomy did not affect ratings for touch pleasantness (*Figure 2a- d* and *Supplementary file 2b*). Regression analysis of brush velocity and VAS scores for all four conditions showed that a negative quadratic regressor provided a better fit than a linear regressor (F test, p=0.001–0.003). The negative quadratic term, $\beta_2$, and extracted

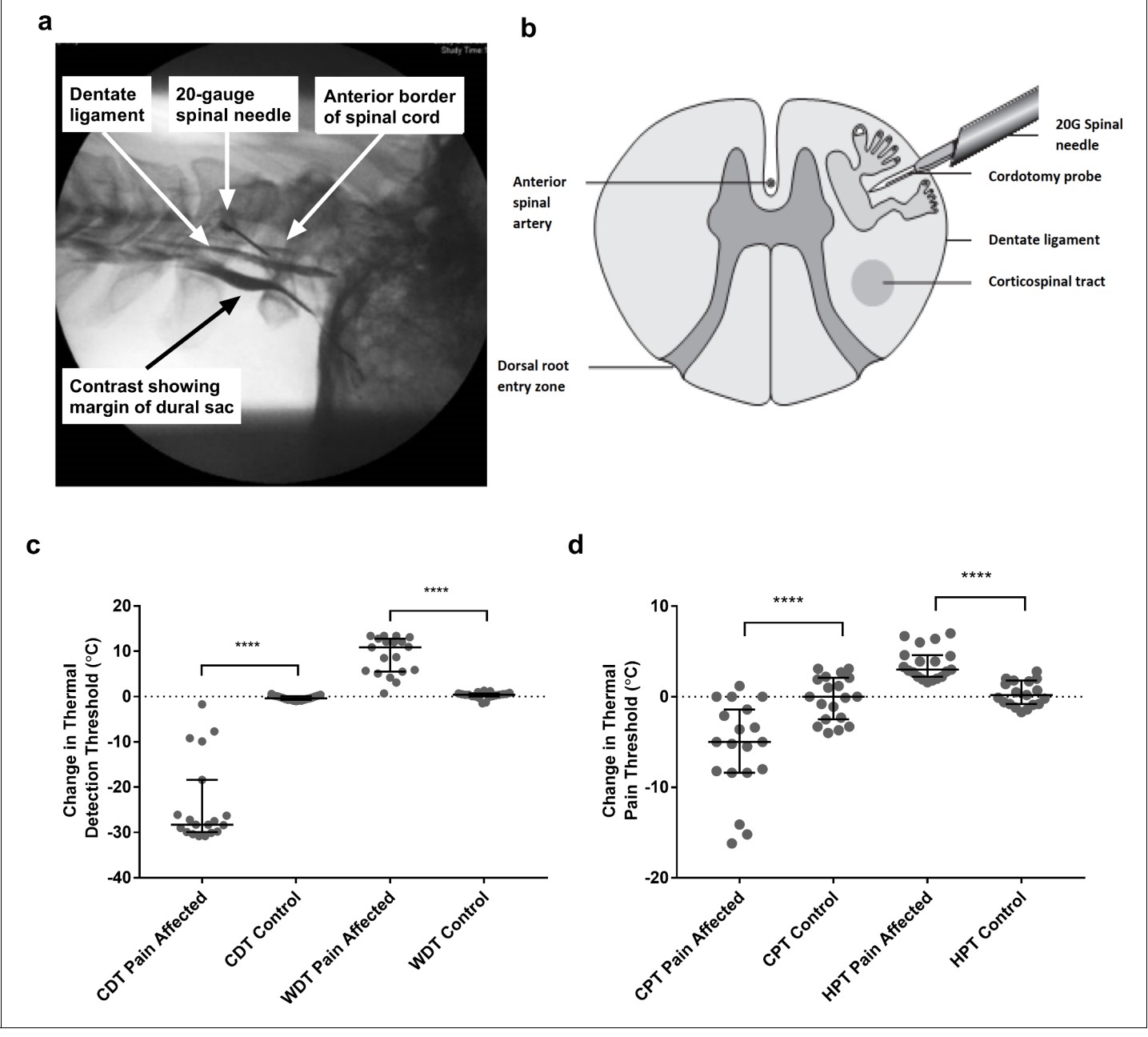

**Figure 1.** Anterolateral cordotomy induces marked deficits in canonical Lamina I spinothalamic tract modalities. Myelogram (**a**) and schematic (**b**) showing the anterolateral cordotomy procedure. Following dural puncture contrast is injected to document the position of the dentate ligament. Radiofrequency lesions are given through the cordotomy probe within the anterolateral funiculus. Dot plots showing changes in pre-cordotomy to post-cordotomy thermal detection and pain thresholds are shown in (**c**) and (**d**) respectively. Data are presented as median and interquartile range. Significant differences (Related-Samples Wilcoxon Signed Rank Test) between the pain affected and control sides are marked with asterisks and show ****p<0.0005. Abbreviations: CDT, Cold Detection Threshold; WDT, Warm Detection Threshold CPT, Cold Pain Threshold; HPT, Heat Pain Threshold.

Y-intercept values for individual patients, which provide measures of the degree of the inverted U-shape and overall perceived touch pleasantness across all velocities respectively, were not significantly altered by cordotomy (*Figure 2d-f* and Supplementary Table 3). For individual patients a negative quadratic regressor provided a better fit than a linear regressor (F test, p=0.05 or less) for 14/19 (pre-cordotomy pain affected, pre-cordotomy control) and 13/19 (post-cordotomy pain affected and post-cordotomy control) patients.

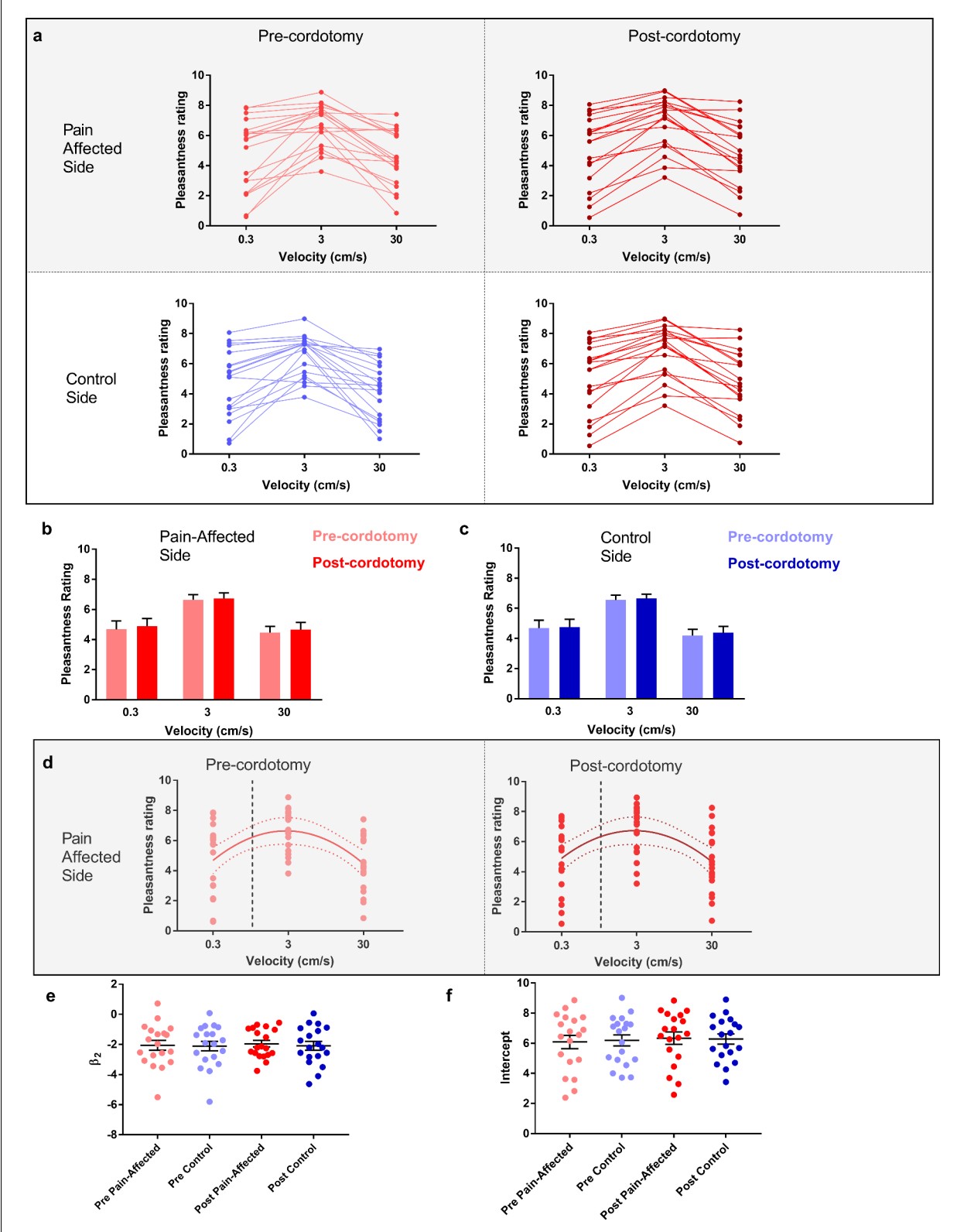

**Figure 2.** The preference for C-Tactile targeted touch and overall touch pleasantness are unaffected by anterolateral cordotomy. (**a**) Raw touch pleasantness rating data for the pain-affected and control sides in the pre-cordotomy as well as post-cordotomy states. Group data for the pain affected side pre-cordotomy and post-cordotomy are shown in (**b**). Group data for the control side pre-cordotomy and post-cordotomy are shown in (**c**). Ratings of touch pleasantness are not significantly affected by anterolateral cordotomy. Dot plots of the mean individual ratings for touch

*Figure 2 continued on next page*

*Figure 2 continued*

pleasantness on the pain-affected side in the pre-cordotomy and post-cordotomy state are shown in (**d**). The lines of best fit with 95% confidence intervals are shown. The vertical dotted line indicates the position of a velocity of 1 cm s-1 on the logarithmic scale. Intercept values were defined as the value of where this 1 cm s-1 line is crossed by the line of best fit. The equations for the fitted curves, $R^2$ as well as F-test results for the pre-cordotomy and post-cordotomy states are ($y = -2.07x^2 + 1.88x + 6.22$; $R^2 = 0.22$; $F_{2, 56} = 7.432$, p=0.001) and ($y = -1.95x^2 + 1.75x + 6.34$; $R^2 = 0.19$; $F_{2, 56} = 6.211$, p=0.004) respectively. F-test results for a linear fit were not statistically significant (p=0.756 and p=0.749 for pre-cordotomy and post-cordotomy states respectively). Dot plots of individual values for $\beta_2$ and intercept pre-cordotomy and post-cordotomy states for both the pain-affected and control side are shown in (**e**) and (**f**). Group data are presented as mean + standard error mean.

Ratings of touch intensity, which show a close correlation with A-β low threshold mechanoreceptor afferent firing rates, increased with increasing brushing velocity as expected (*Figure 3a-c*) (*Löken et al., 2009*). This pattern was present in all conditions, however, VAS touch intensity across all velocities was significantly lower following cordotomy in the pain-affected side (*Figure 3a-b* and *Supplementary file 2b*). Ratings for both touch pleasantness and intensity on the control side (i.e. ipsilateral to the cordotomy lesion) were unaffected by anterolateral cordotomy. Therefore, counter to our prediction that spinothalamic tract lesioning would reduce the pleasant properties of touch, we found instead that touch intensity – a generally accepted discriminative function - was reduced.

We also used the Touch Perception Task (*Guest et al., 2011*; *Ackerley et al., 2014b*) to measure any changes in touch hedonics. In the Touch Perception Task ratings for sensory/discriminative and affective/emotional descriptors are provided in response to specific tactile events. Relative to hairy skin, gentle stroking of skin lacking C-tactile innervation (e.g. palmar glabrous skin) results in lower ratings for positive emotionally relevant terms (e.g. calming and comfortable) (*Ackerley et al., 2014b*; *McGlone et al., 2012*). Here, a stroking stimulus was applied at C-tactile optimal velocity (3 cm s$^{-1}$) using a force-controlled (0.22 N) device attached to which was a material typically perceived as either pleasant (fake fur) or unpleasant (sandpaper). Mean ratings for individual sensory/discriminative and affective/emotional descriptor terms are shown in *Figure 4a*. Using principle component analysis to reduce the number of variables, four sensory/discriminative factors; termed 'texture', 'pile', 'slip' and 'heat'; and three affective emotional factors; termed 'positive', 'arousal' and 'negative'; were extracted from the data sets (see Materials and methods). Each of these factor terms are variably contributed to by the descriptors allowing for computation of an overall weighted factor score. The changes in weighted score for the factor terms between the pre-cordotomy and post-cordotomy states are shown in *Figure 4b-g*. Prior to cordotomy, stroking with fur resulted in high mean descriptor ratings and weighted factor scores for positive emotional terms, as well as discriminative terms relating to surface pile (e.g. fluffy, soft) (*Figure 4a*). However, these were all unaffected by cordotomy, further supporting the finding that following spinothalamic tract disruption the emotional descriptive profile for soft stroking of hairy skin does not shift towards that seen with stimulation of skin lacking C-tactile innervation (*Figure 4a-g*) (*McGlone et al., 2012*). In contrast, for stimulation with sandpaper, which is an unpleasant stimulus, lesioning significantly attenuated roughness perception (*Figure 4a-b*) and, concomitantly, shifted the affective valance of tactile sensation from negative to positive (*Figure 4a,f and g*). Ratings for both sensory and emotional descriptor terms were unaffected on the control side (*Figure 4a-g*).

## Discussion

The development of a velocity-tuned preference to slow touch is dependent on the activity of small diameter afferents, presumably C-tactile fibers. Patients who have congenital C-fiber denervation, but normal A-β fiber function, lack the inverted U-shaped relationship between stroking velocity and pleasantness (*Morrison et al., 2011*; *Macefield et al., 2014*). Instead, their rating patterns indicate a reliance on A-β low threshold mechanosensitive receptor afferent inputs. If there were a dedicated lamina I spinothalamic coding channel responsible for the perception of affective aspects of touch one would expect post-cordotomy affective touch metrics to shift towards those seen in patients with congenital C-fiber denervation. However, here we have shown that, unlike the unambiguous absence of the perceptions of temperature, itch and pain following anterolateral cordotomy, judgments about touch pleasantness, including that predicated on distinctive velocity tuned C-tactile responses, were unaltered. This unexpected finding poses an intriguing question about the

functional neuroanatomy of hedonic touch. How, and in what form, might C-tactile afferents impart their emotionally salient activity on the higher central nervous system?

Slow, stroking stimuli targeting C-LTMR afferents do elicit velocity tuned responses in lamina I projection neurons in rats (*Andrew, 2010*). These projection neurons are, however, wide dynamic range and also respond to noxious stimuli (*Andrew, 2010*). Furthermore, C-LTMR terminals in dorsal horn lamina IIi that connect to lamina I projection neurons (*Lu and Perl, 2003*; *Maxwell et al., 2007*; *Lu et al., 2013*) do so via an interneuronal relay subject to complex regulation (*Larsson and Broman, 2019*). Other recent evidence suggests that rodent C-LTMR afferents access the dorsal column pathway (*Abraira et al., 2017*) via the interneuronal rich dorsal horn zone spanning lamina II_iv through lamina V that receives synapses from myelinated and unmyelinated low threshold mechanosensitive receptor subtypes (*Li et al., 2011*; *Abraira and Ginty, 2013*; *Abraira et al., 2017*). Integrated outputs from this recipient zone target the indirect, post-synaptic dorsal column pathway (*Abraira et al., 2017*). C-LTMR terminals in the dorsal horn paradoxically, given the poor spatial resolution of C-tactile-mediated touch (*Olausson et al., 2002*; *McGlone et al., 2014*; *Olausson et al., 2007*), show precise somatotopic arrangement with little overlap (*Kuehn et al., 2019*). This suggests that C-LTMR afferents, rather than signalling directly, shape the processing of hairy skin A-β subtypes in 'somatotopically relevant' manner (*Kuehn et al., 2019*).

Unlike in individuals with *congenital* small afferent fiber deficits, the integrative processing and shaping between C-Tactile and A-β low threshold mechanoreceptor afferents in neurodevelopmentally intact individuals, whether within the dorsal horn, subcortical regions or distributed cortical

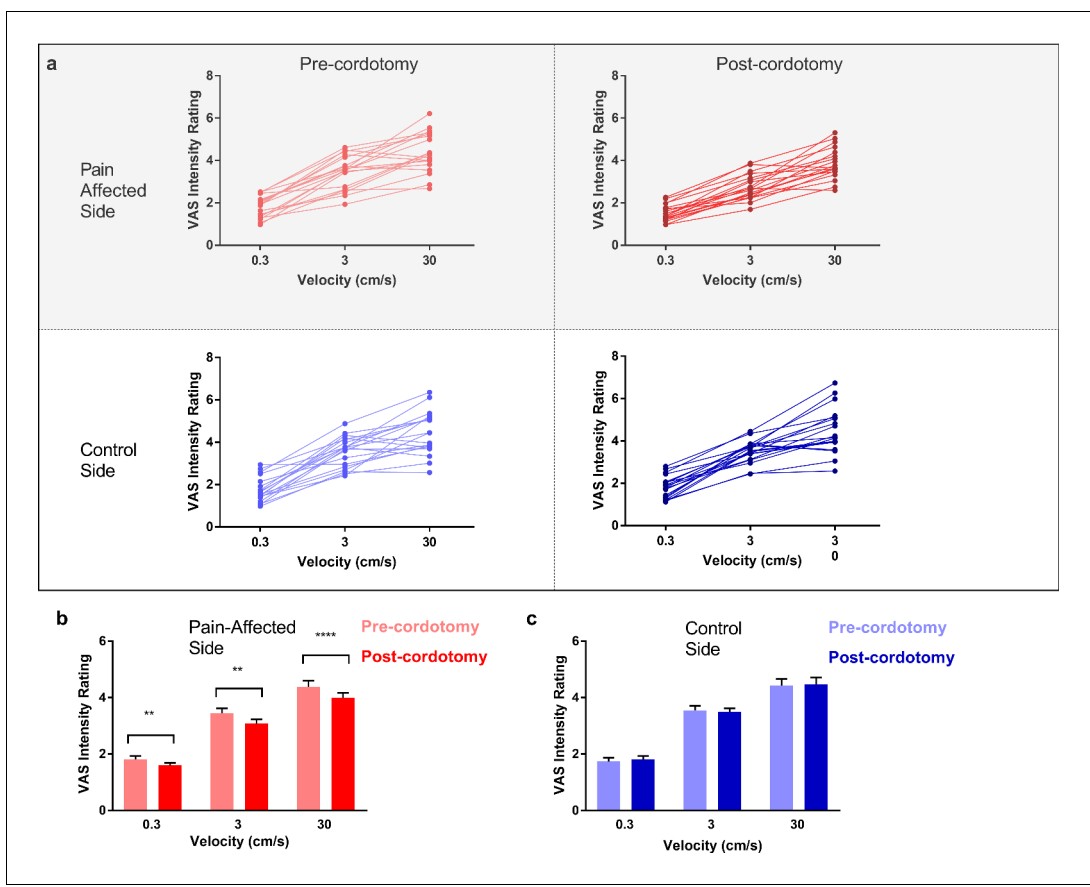

**Figure 3.** Anterolateral cordotomy induces a reduction in perceived touch intensity on the pain-affected side. (a) Raw touch intensity rating data for the pain-affected and control sides in the pre-cordotomy as well as post-cordotomy states. Group touch intensity rating data for the pain affected side pre-cordotomy and post-cordotomy are shown in (b). Group data for the control side pre-cordotomy and post-cordotomy are shown in (c). Group data are presented as mean + standard error mean. Significant differences (Post-hoc analysis) between the pre-cordotomy and post-cordotomy states are marked with asterisks and show **$P_s$ <0.01, **** $P_s$ <0.0005.

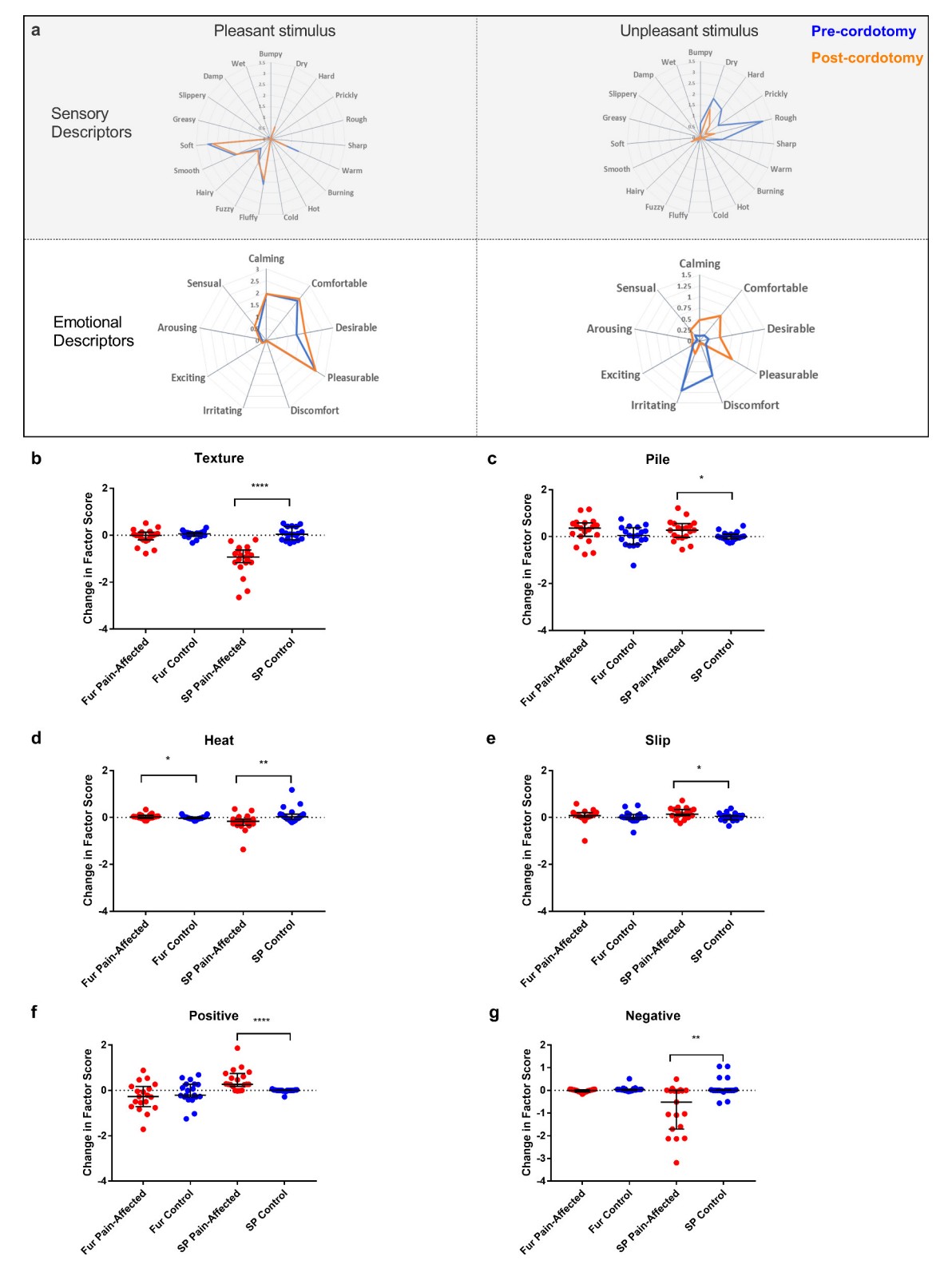

**Figure 4.** Descriptor ratings and factor scores for sensory and emotional terms in the Touch Perception Task. Radar plots showing the mean ratings for sensory and affective descriptor terms in the pre-cordotomy (blue line) and post-cordotomy (orange line) states on the pain affected side are shown in (a). Pleasant and unpleasant touch stimulation was delivered on the forearm using fake fur and sandpaper respectively. Note that the blue and orange lines are almost superimposed for stroking with a pleasant stimulus for both emotional and sensory descriptors. In contrast both sensory and emotional

*Figure 4 continued on next page*

*Figure 4 continued*

descriptor ratings for an unpleasant stimulus are clearly altered by spinothalamic tract lesioning. Markedly lower mean ratings for dry, hard, prickly, rough and sharp are seen post-cordotomy. A clear divergence in the pattern of ratings is seen for emotional descriptors: ratings for negative descriptors are higher than positive descriptors in the pre-cordotomy state but the opposite pattern is seen post-cordotomy. Radar plots for descriptor ratings to stimulation with fur and sandpaper on the control side (not shown) were superimposable for respective pre-cordotomy and post-cordotomy states as well as for the equivalent material in the pre-cordotomy state on the pain affected side. The absolute change in the factor score between the pre-cordotomy and post-cordotomy states for stimulation with fur and sandpaper on the pain-affected and control sides are shown in the dot plots for sensory (b - e) and emotional (f - g) factors. Factor scores for stroking with sandpaper are significantly affected by cordotomy with evidence of a marked reduction in ratings for the texture group (b) and a more modest reduction in ratings for heat terms (d). There are small but significant increases in ratings for descriptor terms in the pile (c) and slip (e) group. Only heat (d) is significantly altered for stimulation with fur. For stroking with an unpleasant stimulus highly significant increases and decreases in emotional factor scores were seen for positive (f) and negative (g) terms respectively. These are unaffected for stroking with fur. No significant change in the emotional factor 'arousal' was seen (data not shown). Bars depicting median and interquartile ranges are shown. Significant differences (Related-Samples Wilcoxon Signed Rank Test) between the pain-affected and control sides are marked with asterisks and show *p<0.05, **p<0.01, ***p<0.001, ****p<0.0005. Abbreviation: SP, sandpaper.

regions, will have been present over a lifetime. Whilst the emerging realization of the complexity of processing of tactile information at the earliest central nervous system relay (*Abraira et al., 2017*) suggests a more complex explanation it is conceivable that the cordotomy patients may have learned to associate certain tactile velocities with touch pleasantness and thus rely purely on ascending A-β low threshold mechanoreceptor afferent inputs when making judgements about the affective/emotional properties of touch. In either case the current findings indicate in individuals with a neurodevelopmentally normal somatosensory system that fibers ascending outside the anterolateral funiculus, most likely within the dorsal columns, provide sufficient information to conserve judgements about touch pleasantness.

A contralateral reduction in ratings for the intensity of stroking touch across all velocities was seen following anterolateral cordotomy although monofilament tactile detection thresholds were not affected. It is unlikely that these effects relate to a selective loss of ascending C-tactile inputs. There is evidence that C-tactile afferents contribute to the relative preservation of monofilament tactile detection in A-β denervated individuals and under conditions of A-fiber blockade (*Cole et al., 2006*; *Nagi et al., 2015*). However, their mean firing rates, unlike those of A-beta afferent LTMRs, do not correlate with the touch intensity ratings that increase in parallel stroking velocity (*Löken et al., 2009*; *Ackerley et al., 2014a*). Although the precise mechanisms underlying the reduction in the perceived intensity of stroking touch following cordotomy are unclear they too may relate to the distributed signalling of tactile information across multiple ascending pathways.

The current findings support such an integrative model of hedonic touch also for human hairy skin. They are, in fact, incompatible with a segregated model of touch where emotional and discriminative elements are signaled in anatomically discrete second order pathways. Indeed, the contralateral attenuations of texture perception and touch intensity seen post-cordotomy indicate that, for hairy skin, tactile information quintessentially regarded as discriminative and dependent on A-β activity (*Saal and Bensmaia, 2014*; *Manfredi et al., 2014*; *Lieber et al., 2017*), also partly relays in crossed pathways ascending the anterolateral funiculus.

## Materials and methods

### Participants

Twenty patients were recruited in accordance with the Health Research Authority National Research Ethics Service (study reference 14/NW/1247). The study was conducted in accordance with the Declaration of Helsinki. All patients were admitted to the Walton Centre, Liverpool, UK and suffered from intractable unilateral cancer-related pain below the cervical level C4 with an expected lifespan of less than 12 months. It was not possible to test one patient in the post-operative state. Of the 19 patients nine were female. The patients' demographic and clinical details are shown in *Supplementary file 1*. No patient had pre-existing symptoms or signs of neurological impairment, including pain, in the region of sensory testing. All patients were medicated with regular and *pro re nata* opioids as well as a variety of non-opioid analgesia. The median and range for numeric rating scale of average 4 hr pain, maximum pain in the past 4 hr and current pain were 76 (20-90), 98 (79-

100) and 50 (10-81) respectively. A large number (13/19) of patients had previously received chemotherapy with potential peripheral neurotoxicity although no patient described ongoing symptoms potentially attributable to this.

Opioid treatment (*Martel et al., 1995*; *Case et al., 2016*), chronic pain (*Case et al., 2016*) and chemotherapy-induced neurotoxicity (*Geber et al., 2013*; *Krøigård et al., 2014*) could all, in principle, impact on sensory testing. However, pre-procedural thermal and thermal pain detection thresholds were normal in the area of sensory testing and there was no pre-procedural evidence of impaired sensory discriminative or affective touch (see main article). Furthermore, since the study paradigm compared lesioned versus non-lesioned sides and pre-versus post-lesion states one would expect a right-left or pre-post difference in measures of affective or discriminative touch to be detected even if there was an underlying subtle (drug or pain induced) baseline 'abnormality' in the function or processing of C-tactile afferents or a generalized procedural effect.

## Spinothalamic tract ablation

Antero-lateral cordotomy (*Bain et al., 2013*) was performed at the cervical level C1/C2 contralateral to the cancer-related pain. The procedure was performed with sedation and local anesthesia. Following dural puncture with a 20G spinal needle the cordotomy electrode was advanced into the antero-lateral quadrant of the spinal cord (*Figure 1*). Positioning in the spinothalamic tract was verified by eliciting cold, heat or other painful sensations, encompassing the region of cancer-related pain, using 50 Hz electrical stimulation through the cordotomy electrode. Motor twitch threshold using 10 Hz stimulation was also performed to assess proximity to the corticospinal tract. Adjustments of the electrode were made to maximize location with the spinothalamic tract and minimize proximity to motor pathways. The spinothalamic tract was disrupted using a radiofrequency current which produces a heat-induced lesion. This was performed in steps, typically starting at 65°C for 25–30 s, with a maximum temperature of 85°C. Lesioning of the spinothalamic tract was confirmed in the operating theatre by demonstrating a contralateral loss of temperature sensation on clinical examination. Operative details for all cases are shown in *Supplementary file 1*.

## Experimental design

All patients underwent pre-procedure testing, either on the morning of or day before cordotomy. Post-cordotomy testing was undertaken at least four hours following the procedure to allow for recovery from operative sedation. All post-cordotomy assessments were performed within 72 hr of the procedure, when spinothalamic deficits are likely to be maximal. Pre-procedure and post-procedure testing lasted approximately 90 min. All assessments were performed on the dorsal aspect of both the right and left forearm. The order of testing with respect to right and left was randomized.

## Pleasant touch

Assessment of gentle dynamic touch was made using a 70 mm goat's hair artist brush. Patients were prevented from seeing the tested extremity throughout the experiment. Stimuli were delivered manually in a proximal to distal direction over a 10 cm distance marked on the forearm at velocities of 0.3, 3 and 30 cm s$^{-1}$, chosen to reflect C-tactile optimal (3 cm s$^{-1}$) and sub-optimal (0.3 and 30 cm s$^{-1}$) stimuli. A computerized visual meter was used during training and testing sessions. Six stimuli at each velocity were given on each side in a computer-generated pseudorandom order. An inter-stimulus interval of at least 10 s was allowed to prevent fatigue in C-tactile firing. After each stroke patients rated both the pleasantness and intensity of the stimulation using a 20 cm paper visual analogue scale. Anchor points for touch intensity were no sensation (0) and very intense (10). For pleasantness anchor points were 'unpleasant' (−10) and 'pleasant' (10) with 0 representing a neutral stimulus.

## Tactile acuity and graphesthesia

Mechanical detection thresholds were determined using von Frey monofilaments (Optihair2- Set Nervtest, Germany) according to the 'method of limits' (*Rolke et al., 2006*). Two-point discrimination (TPD) was determined using mechanical sliding calipers. Five ascending and descending assessments, centred around the subject's TPD threshold, were conducted. The geometric mean of the obtained values was calculated for the threshold. Graphesthesia was used as a test of dorsal column

function (*Bender et al., 1982*). Participants were asked to identify numbers 3, 4 and 5 that were drawn on the skin, approximately 6 cm in top-bottom dimension, using the blunt end of a Neurotip (Owe Mumford Ltd, UK). Initially testing was performed with the eyes open to ensure that the task was understood. Each number was presented three times in a pseudorandom order with eyes closed.

## Thermal threshold testing

Innocuous cold and warm detection as well as cold and heat pain thresholds were measured using the method of limits with the MEDOC TSA II (Medoc, Ramat Yishai, Israel). The thermode had a surface area of 9.0 cm$^2$ and baseline temperature of 32°C. Thresholds were obtained using ramped stimuli of 1 °C s$^{-1}$, the patient terminating the ramp with a button press. The mean of three consecutive temperature thresholds was calculated. The maximum and minimum limit of the thermode was 50°C and 0°C. Once the maximum or minimum temperature had been attained the temperature of the thermode immediately started to return toward baseline.

## Pinprick testing

Assessment of pinprick sensation was made using a Neurotip (Owe Mumford Ltd, UK).

## Itch

Assessment of itch sensation was made using cowhage. Cowhage spicules contain the pruritogen mucunain (*Reddy et al., 2008*; *Davidson and Giesler, 2010*) and on skin contact induce a histamine independent itch via activation of proteinase-activated receptors-2 and −4 (*Reddy et al., 2008*; *Davidson and Giesler, 2010*). Recordings in primates have shown that cutaneous application of cowhage activates ascending spinothalamic projection neurons (*Davidson et al., 2012*). Approximately 20 cowhage spicules were collected onto a cotton bud and rubbed directly on a 1cm2 skin site for 20 s. Spicules were then immediately removed with a strip of lightly-adhesive paper tape (Micropore, 3M, USA). Assessments were made post-cordotomy only. Patients rated the intensity of itch on a numeric rating scale (0–100). If no perception of itch was elicited cowhage application was repeated up to a maximum of three times before the sensation was judged to be absent.

## The touch perception task

The Touch Perception Task was developed as a validated descriptive scale for touch perception (*Guest et al., 2011*). The full Touch Perception Task consists of 26 sensory and 14 emotional descriptors that provide information about differing aspects of touch in relation to specific tactile stimulations. A shortened form consisting of 28 descriptors was administered omitting seven sensory (firm, gritty, jagged, lumpy, rubbery, sticky and vibrating) and five emotional (sexy, thrilling, enjoyable, soothing and relaxing) descriptors (*Supplementary file 3a*). Stimuli were administered using a manual tactile stimulator that delivers a force-controlled stimulus at 0.22N. To this either sandpaper (grade: P120, average particle diameter 120 μm) or artificial fur (soft 10 mm long hairs, average diameter approximately 50 μm) were attached with an application dimension of 80 × 50 mm. Artificial fur and sandpaper have been used previously to provide extremes of tactile stimuli (*Ackerley et al., 2014b*). The manual tactile stimulator was moved over the skin at 3 cm s$^{-1}$ over a 10 cm distance in a proximal to distal direction. The order of testing with respect to the type of material was randomized.

## Sample-size estimation

Using an F-test power calculator for repeated measures ANOVA, assuming correlation among repeated measures for pleasantness ratings (primary outcome) of 0.7, for significance level of 0.05 twenty participants would grant approximately 80% power for an effect size *f* of 0.25 or 90% power for an effect size *f* of 0.4. These are conservative estimates. Previous studies comparing individuals with Hereditary Sensory and Autonomic Neuropathy type V (a mix of heterozygous and homozygous carriers) to healthy controls have detected highly significant differences with ten participants per group (*Morrison et al., 2011*).

## Data analysis

Statistical analyses were carried out with SPSS (version 23; IBM, Armonk, NY), Excel 2010 (Microsoft TM) and Graphpad Prism (version 7.04; GraphPad Software, La Jolla, CA). Rating data for pleasantness and intensity were averaged for each participant and each velocity and these average values were used in the reported analysis of variance (ANOVA).

Regression analysis was performed to assess the shape of rating curves. Using logarithm-transformed values for the independent variable, 'velocity', rating data were entered into the regression model as both linear and quadratic terms. Analysis was performed on both a group level, using average rating scores, and individually, using all individual rating scores, to extract quadratic term and intercept values (*Morrison et al., 2011*). These values describe the two key components of typical pleasantness ratings to gentle dynamic touch in healthy individuals: the degree of the inverted U-shape provides a measure of the velocity-dependent preference for C-tactile targeted touch, whereas, the intercept value reflects overall perceived touch pleasantness across all velocities. Quadratic terms that are more negative represent a greater preference to C-tactile targeted velocities when compared to fast and very slow touch. Intercept values that are higher reflect higher pleasantness ratings encompassing all velocities.

As the study population was substantially older than in previous studies and because an abbreviated version of the Touch Perception Task was used, a factor analysis using information obtained in the pre-cordotomy state and healthy control participants was performed to reduce the number of variables into fewer numbers of factors. Scores from sensory and emotional descriptors were entered in separate factor analyses to yield sensory and emotional factors respectively. The approach was similar to that used in previous studies.

Four factors, termed 'texture', 'pile', 'slip' and 'heat' which explained 39.5%, 14.0%, 11.6% and 8.1% of the total variance respectively were extracted from the sensory descriptor terms. Three factors, termed 'positive', 'arousal' and 'negative' which explained 65.6%, 14.8% and 8.1% of the total variance were extracted from the emotional descriptor terms. These findings are broadly consistent with previous investigation (*Guest et al., 2011*; *Ackerley et al., 2014b*). Factor loadings (regression and correlation coefficients) for significantly contributing descriptors are presented in order of magnitude along with the variance and covariance incorporated in each factor in *Supplementary file 3b and c*. A factor weight matrix was then used to compute overall factor scores for each sensory and emotional factor. These were subsequently used to explore differences following cordotomy.

Repeated measures ANOVA was used to explore significant differences in pleasantness and intensity rating data, intercept and quadratic terms as well as mechanical detection and two-point discrimination thresholds. All models had factors of time (pre- and post-cordotomy) and side (pain-affected and control). A third factor of either velocity (0.3, 3 and 30 cm s$^{-1}$) or material (fur and sandpaper) were used when appropriate. Data were logarithm transformed when appropriate (Shapiro-Wilk's test of normality p<0.05). In the case of outliers, assessed as a value that fell 3 times above or below the bounds of the interquartile range, analyses were repeated after removal. All analyses were robust to outlier removal (*$\beta_2$: pre-cordotomy pain-affected - 1 point of 19 points; $\beta_2$: pre-cordotomy control – 1 point of 19 points; Mechanical Detection Threshold: post-cordotomy pain-affected – 1 point of 19 points; Pleasantness ratings: post-cordotomy pain-affected – 1 point of 19 points*). Significant interaction effects were followed up using simple main effects and pairwise comparisons with Sidak's correction (denoted in the text as Ps). F approximations to Pillai's trace are reported. Wilcoxon signed rank test was used to explore pre- and post-cordotomy as well as pain affected versus control side differences in non-parametric distributed data. Statistical significances were sought at the p<0.05 level.

## Acknowledgements

We thank The Walton Center clinical staff for their assistance with recruitment and for facilitating assessment of participants. We thank Rochelle Ackerley for advice regarding the Touch Perception Task. This work was supported by the Pain Relief Foundation (AGM and FM).

## Additional information

### Funding

| Funder | Author |
|--------|--------|
| Pain Relief Foundation | Andrew G Marshall<br>Francis P McGlone |

The funders had no role in study design, data collection and interpretation, or the decision to submit the work for publication.

### Author contributions

Andrew G Marshall, Conceptualization, Data curation, Formal analysis, Funding acquisition, Validation, Investigation, Methodology, Project administration; Manohar L Sharma, Conceptualization, Resources, Investigation, Methodology; Kate Marley, Investigation; Hakan Olausson, Conceptualization, Supervision; Francis P McGlone, Conceptualization, Resources, Supervision, Funding acquisition, Methodology, Project administration

### Author ORCIDs

Andrew G Marshall  https://orcid.org/0000-0001-8273-7089

### Ethics

Human subjects: Ethical approval was obtained through the Health Research Authority National Research Ethics Service (Preston NRES committee, study reference 14/NW/1247). The study was conducted in accordance with the Declaration of Helsinki. Written informed consent was taken from all study participants.

### Decision letter and Author response

Decision letter https://doi.org/10.7554/eLife.51642.sa1
Author response https://doi.org/10.7554/eLife.51642.sa2

## Additional files

### Supplementary files

• Supplementary file 1. Demographic and clinical data of patients undergoing anterolateral cordotomy. Abbreviations: NRS, Numeric rating score 0–100. Age is displayed as a range to limit indirect identifiers.

• Supplementary file 2. Statistics summary tables for thermal sensation and touch. (**a**) Summary of thermal threshold and discriminative touch sensation testing in the pre-cordotomy and post-cordotomy states. Significant differences (Related-Samples Wilcoxon Signed Rank Test) be-tween the pre-cordotomy and post-cordotomy states are marked with asterisks and show ****p<0.0005. Abbreviations: CDT, Cold Detection Threshold; WDT, Warm Detection Threshold; CPT, Cold Pain Threshold; HPT, Heat Pain Threshold; MDT, Mechanical Detection Threshold; TPD, Two-Point Discrimination; NRS, Numeric Rating Scale; IQR, Interquartile Range; SD, Standard Deviation. (**b**) Summary of three-way repeated measure ANOVA for the effects of velocity, side (control versus pain affected) and time (pre-cordotomy versus post-cordotomy) on pleasantness ratings, intensity ratings, negative quadratic term and intercept.

• Supplementary file 3. Touch Perception Task analysis. (**a**) List of sensory and emotional descriptors used in the Touch Perception Task. (**b**) Sensory descriptors factor analysis Three significant factors were found in the emotional descriptors data (those contributing >5% of the variance; detailed in the Materials and methods) and named Texture, Pile, Heat and Slip. The descriptors and their significant loadings (>0.3) are shown for both the regression (pattern matrix) and the correlation (structure matrix) factor analysis output. (**c**) Emotional descriptors factor analysis Three significant factors were found in the emotional descriptors data (those contributing >5% of the variance; detailed in the Materials and methods) and named Positive Affect, Arousal and Negative Affect. The descriptors

and their significant loadings (>0.3) are shown for both the regression (pattern matrix) and the correlation (structure matrix) factor analysis output.

- Transparent reporting form

### Data availability

All data generated or analysed during this study are either included in the manuscript and supporting files or available on Open Science Framework (https://osf.io/g8vyk/).

The following dataset was generated:

| Author(s) | Year | Dataset title | Dataset URL | Database and Identifier |
|---|---|---|---|---|
| Marshall AG | 2019 | Marshall_September_2019_ Cordotomy_Database_ Psychophysics | https://osf.io/g8vyk/ | Open Science Framework, g8vyk |

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
