## [Decision Letter]

**Acceptance summary:**

This is an important piece of work, elegant in its simplicity and match of the methods to the question. Now we know that C-fiber mediated pleasurable touch requires a pathway outside of the STT.

**Decision letter after peer review:**

Thank you for submitting your article "Spinal signaling of C-fiber mediated pleasant touch in humans" for consideration by *eLife*. Your article has been reviewed by three peer reviewers, including Peggy Mason as the Reviewing Editor and Reviewer #1, and the evaluation has been overseen by Christian Büchel as the Senior Editor. The following individuals involved in review of your submission have agreed to reveal their identity: David A Mahns (Reviewer #2); Martin Schmelz (Reviewer #3).

All reviewers agree that this work is important and compelling. The changes suggested can all be addressed through revising the writing.

Please address the issues raised by reviewer 3; viz the roles of Ab learning and of a decrease in pain allowing for pleasantness.

Reviewer #1:

This is a clear and important demonstration that C-fibers underlying pleasant touch perception do not travel exclusively or even predominantly in the STT. The experiment is delightfully simple as in straightforward. Results are clear. Conclusions are warranted.

"touch intensity – a generally accepted discriminative function – was reduced," but only on pain affected side. This is not a critical point but warrants at least a mention of "we can’t explain this" or "we speculate that…."

I did not understand Figure 4B-G. Are these different factors that each depend on a group of words relevant to the term listed in the figure? This is a short paper and the authors have plenty of room to expand. I suggest taking a sentence or two to bring the reader along.

Reviewer #2:

This article addresses a simple question in somatosensory physiology, which ascending spinal pathways relay information about pleasant touch in patients undergoing an anterolateral cordotomy. The authors have used patients scheduled for cordotomy to test whether the perception of pleasant touch evoked by gentle brushing, a stimulus presumed to be preferential for C-tactile fibres, is abolished by sectioning the contralateral tract at the C1-2 level of the spinal cord. Prior to sectioning patients felt spontaneous / ongoing pain in thoracic, lumbar and sacral innervation territories whereas itch, touch, pain and thermal sensibility were normal at the test site in the forearm. Following cordotomy, there was an amelioration of clinical pain, while itch and touch (mechanical)-evoked pain were abolished, cold sensation / cold pain detection thresholds were fell and warm detection/ heat pain threshold rose. In contrast the pleasantness ratings were unaffected by chordotomy and retained an inverted U tuning frequently ascribed to response profile of C-tactile fibres and pleasant touch.

Overall the article is clear and concise and raises a key question, is information about pleasant touch relayed in dedicated pathways or distributed across multiple ascending pathways. The results clearly demonstrate that, in contrast to the prevailing narrative in the field of affective touch, that it is indeed the latter – the authors are commended for presenting an alternative view that will drive a deeper investigation of this field. Consequently, the results are clear novel and should be published as soon as possible.

Reviewer #3:

In this study patients undergoing antero-lateral cordotomy were investigated prospectively concerning the expected reduction of small fiber mediated function as compared to expectedly unchanged A-β function. In particular low threshold mechanosensitive C-fibers (C-touch) were studied. The main result is given by the unchanged pleasantness ratings after cordotomy contrasting the basically abolished other small fiber functions such as pain and itch. Moreover, touch intensity ratings were reduced indicating reduced processing of A-β information.

The authors conclude that C-touch fiber function is not abolished by the antero-lateral cordotomy and therefore might rather follow the dorsal column pathway in human. Moreover, they propose an integrative model of touch in which c-touch fibers, albeit primarily involved in emotional aspects of touch, also contribute to touch discrimination by their spinal interaction with A-β input.

The beauty of this paper is investigating patients prospectively before a defined neurosurgical procedure to allow for conclusions on the effects on central pathway of the studied sensory qualities directly in humans. In particular, this approach allows to study the question whether or not the C-touch fibers are dependent on the spinothalamic tract. While the approach is straight forward some additional aspects need to be dealt with in the Discussion to strengthen the implications of unchanged pleasantness ratings after anterolateral chordotomy:

The authors suggest that unchanged velocity-tuned pleasantness ratings imply intact C-touch fibers as this function is abolished in patients with congenital C-fiber denervation. However, it is unclear to which extent live-long learning processes might be sufficient to link a certain touch velocity (as detected by the A-beta system) to pleasantness. Such a learning process might reduce the impact of actual sensory input from C-touch fibers.

Moreover, it is unclear to which extent abolished pain from the test sites affected pleasantness ratings.

It is unclear how many patients suffered from brush evoked allodynia before the operation. The observed reduction of touch intensity after chordotomy might be linked to possibly abolished allodynia or reduced ongoing nociceptive input.

The authors propose an integrative view in which C-touch fibers contribute to discriminative touch perception. It is not quite clear how they link this probably valid assumption to their results: reduced touch perception would not quite fit to such an interaction on the spinal level as this connection can be assumed to be unaffected by the chordotomy.

---

## [Author Response]

Reviewer #1:This is a clear and important demonstration that C-fibers underlying pleasant touch perception do not travel exclusively or even predominantly in the STT. The experiment is delightfully simple as in straightforward. Results are clear. Conclusions are warranted."touch intensity – a generally accepted discriminative function – was reduced," but only on pain affected side. This is not a critical point but warrants at least a mention of "we can’t explain this" or "we speculate that…."

The following has been added to the Discussion:

“A contralateral reduction in ratings for the intensity of stroking touch across all velocities was seen following anterolateral cordotomy although monofilament tactile detection thresholds were not affected. […] Although the precise mechanisms underlying the reduction in the perceived intensity of stroking touch following cordotomy are unclear they too may relate to the distributed signalling of tactile information across multiple ascending pathways.”

I did not understand Figure 4B-G. Are these different factors that each depend on a group of words relevant to the term listed in the figure? This is a short paper and the authors have plenty of room to expand. I suggest taking a sentence or two to bring the reader along.

The following has been added:

“Ratings for individual descriptor terms and weighted scores for factors, extracted using principle component analysis (see Materials and methods), were calculated.”

Has been replaced by:

“Here, a stroking stimulus was applied at C-tactile optimal velocity (3 cm s^-1^) using a force-controlled (0.22 N) device attached to which was a material typically perceived as either pleasant (fake fur) or unpleasant (sandpaper). […] The changes in weighted score for the factor terms between the pre-cordotomy and post-cordotomy states are shown in Figure 4B-G.”

Reviewer #3:[…] The beauty of this paper is investigating patients prospectively before a defined neurosurgical procedure to allow for conclusions on the effects on central pathway of the studied sensory qualities directly in humans. In particular, this approach allows to study the question whether or not the C-touch fibers are dependent on the spinothalamic tract. While the approach is straight forward some additional aspects need to be dealt with in the Discussion to strengthen the implications of unchanged pleasantness ratings after anterolateral chordotomy:The authors suggest that unchanged velocity-tuned pleasantness ratings imply intact C-touch fibers as this function is abolished in patients with congenital C-fiber denervation. However, it is unclear to which extent live-long learning processes might be sufficient to link a certain touch velocity (as detected by the A-beta system) to pleasantness. Such a learning process might reduce the impact of actual sensory input from C-touch fibers.

The perception of touch will depend on the entirety of tactile afferent inputs. In a neurodevelopmentally intact somatosensory system there will almost certainly be considerable integration of A-β low threshold mechanosensitive receptor and C-tactile afferents at a number of anatomical levels. The authors agree that individuals may 'learn' to associate touch with a positive affective valence (engendered by C-tactile afferents inputs) with a particular velocity that could be signaled and detected by A-β low threshold mechanosensitive receptor fibres. However, this does not change the conclusion that with an intact somatosensory system fibres ascending outside the spinothalamic tract are sufficient to preserve the perception of touch pleasantness. The following has been added to the Discussion:

“Unlike in individuals with *congenital* small afferent fibre deficits, the integrative processing and shaping between C-Tactile and A-β low threshold mechanoreceptor afferents in neurodevelopmentally intact individuals, whether within the dorsal horn, subcortical regions or distributed cortical regions, will have been present over a lifetime. […] In either case the current findings indicate in individuals with a neurodevelopmentally normal somatosensory system that fibres ascending outside the anterolateral funiculus, most likely within the dorsal columns, provide sufficient information to conserve judgements about touch pleasantness.”

Moreover, it is unclear to which extent abolished pain from the test sites affected pleasantness ratings.

The patients had no pain at the test sites. This has been further emphasised in the

Methods

“No patient had pre-existing symptoms or signs of neurological impairment,

including pain, in the region of sensory testing.”

and Results

“All sensory testing was performed on hairy skin of the dorsal forearm distant to

the sites of clinical pain. No patient had pre-existing neurological deficits in the

area of testing.”

It is unclear how many patients suffered from brush evoked allodynia before the operation. The observed reduction of touch intensity after chordotomy might be linked to possibly abolished allodynia or reduced ongoing nociceptive input.

No patient had allodynia at the test site and indeed had ‘normal’ pleasantness ratings pre-cordotomy on both the painful and control sides (see Results). A single patient complained of allodynia on the chest wall which is distant from the test site (see Supplementary file 1).

The authors propose an integrative view in which C-touch fibers contribute to discriminative touch perception. It is not quite clear how they link this probably valid assumption to their results: reduced touch perception would not quite fit to such an interaction on the spinal level as this connection can be assumed to be unaffected by the chordotomy.

Rather than suggesting that C-tactile fibres contribute to tactile discrimination we are proposing that projection neurons in the anterolateral funiculus contribute to aspects of touch that are typically thought of as discriminative. Since the encoding of the intensity of an innocuous stroking cutaneous stimulus and the signaling of texture are thought to be A-β mediated this implies that A-β inputs gain access to ascending pathways in the anterolateral funiculus.

The following statement about touch intensity has been added to the Discussion:

“A contralateral reduction in ratings for the intensity of stroking touch across all velocities was seen following anterolateral cordotomy although monofilament tactile detection thresholds were not affected. […] Although the precise mechanisms underlying the reduction in the perceived intensity of stroking touch following cordotomy are unclear they too may relate to the distributed signalling of tactile information across multiple ascending pathways.”

Also, to reflect the findings that suggest projection neurons in the anterolateral funiculus contribute to aspects of touch that are typically thought of as discriminative the Discussion ends with the statement:

“Indeed, the contralateral attenuations of texture perception and touch intensity seen post-cordotomy indicate that, for hairy skin, tactile information quintessentially regarded as discriminative and dependent on A-β activity (Saal and Bensmaia, 2014; Manfredi et al., 2014; Lieber et al., 2017), also partly relays in crossed pathways ascending the anterolateral funiculus.”